# OpenReview forum: "MEMORIES THAT DISCRIMINATE: DETECTING AND CORRECTING BIAS IN PERSONALIZED HIRING AGENTS"
_ICLR.cc/2026/Workshop/AFAA — AFAA 2026 Poster_

### Official Review · Reviewer_yvMm · 2026-02-21
**Diagnosing and mitigating bias propagation in memory-present personalized hiring agents**

**Rating:** 4
**Confidence:** 3

**Summary:**

This paper talks about the how memory bias persists in agents, using a recruitment agent as an example.  The recruiter agent uses the past recruitment behavior as the memory ) to personalize the job description, retrieve a ranked candidate list, and rerank candidates when required, thus creating multiple stages where biased memory interpretation can affect decisions. Then it also introduces 2 strategies FRL and TA-ICFRL.

**Strengths:**

- This is a hot topic right now, so well timed and an important problem solving.
- Clear delineation of stages where bias can be introduced. Decomposition of the agent into clear stages.
- Introduces 2 novel strategy to nudge the mitigation of bias.

**Weaknesses:**

- talks only about gender bias, which is insufficient. Need to introduce more bias types here to make the paper strong.
- The strategies provided are prompt based, but not guaranteed compliance by LLMs.
- No mention of temperature of the models used

---

### Official Review · Reviewer_fFFz · 2026-02-22

**Rating:** 3
**Confidence:** 4

**Summary:**

This paper investigates bias in memory-enhanced LLM-based hiring agents, demonstrating that personalization introduces and amplifies gender bias across retrieval, query generation, and re-ranking stages. The authors propose FRL and TA-ICFRL as prompt-level mitigation strategies.

**Strengths:**

Timely and underexplored problem at the intersection of agentic AI and fairness. Systematic experimental design with seven configurations isolating bias at different workflow stages. Evaluation across multiple LLMs strengthens generalizability. The proposed mitigations are practical for black-box API settings.

**Weaknesses:**

Evaluation is limited to gender bias only; intersectional analysis is absent despite being discussed. The Cumulative Group Attention metric, while intuitive, lacks theoretical grounding. TA-ICFRL reduces but doesn't eliminate bias, and improvements over FRL are modest. The recruitment scenario is synthetic, limiting ecological validity. The evaluation is confined to a single sensitive attribute and one synthetic domain. The metrics, while reasonable, are novel without formal validation. The paper also lacks a discussion of when personalization-driven bias might be acceptable versus harmful, which weakens the conceptual framing.

**Limitations**: Single-domain evaluation, no multi-turn interaction analysis, and prompt-based mitigations may be fragile across prompt variations.

---

### Official Review · Reviewer_hzmP · 2026-02-22
**Bias can arise in memory Enhanced LLM Agents especially in Recruiting domain.**

**Rating:** 4
**Confidence:** 4

**Summary:**

The authors propose the measurement of biases that may be induced in memory enhanced LLM agents due to inherent biases in the data and through their interpretations of this data as memory. When tasks are performed by such agents such as selecting candidates for a job, ranking them etc, the preferences of the recruiter/user of the agents can influence this selection. The paper also comes up with mitigation strategies primarily involving prompt augmentation that work well with agents using LLMs via API call and in situations where whitebox access to LLM internals is not feasible.

**Strengths:**

The use of persistent and longer term memory are becoming more mainstream in agents as a means to be able to accomplish long horizon tasks and also to incorporate user preferences for these tasks. The paper attempts to study the biases that may be introduced with the inclusion of these memory elements and how prompt based mitigations can be applied for reducing the effects of such biases.

The paper sets up an experimental setup in the recruiting domain where tasks for agents such as retrieval of candidates based on raw query (baseline), hiring history memory based query, job description creation and personalized retrieval and ranking are conducted using the Bias in Bios dataset by De-Artega et Al (2019). A total of 7 research questions are set up to see where bias creeps in from baseline retrieval or through re-ranking by agent with variations such as explicit balanced retrieval performed as well. The metrics used are cumulative group attention for measuring the impact of ranking of candidates and meritocratic unfairness inspired by meritocratic fairness.

Towards the end, solutions such as Fairness Regulation Learning (FRL) and Task Aware in Context Fairness Regulation Learning are proposed and evaluated on the same experimental setup which shows improvement of 53% and 60% in meritocratic unfairness respectively.

**Weaknesses:**

This paper primarily focuses on the tasks of hiring and recruitment for its study and experiments however the methods do seem general and extendable to other domains perhaps as future work.

The hiring agent’s experimental setup can be described better and it lacks details such as which model is being used to primarily drive the agent. The authors have provided prompts and templates in the appendix. They should also consider providing the experiment’s code repo for reproducibility.

The mitigation methods rely on the prompt augmentation which ultimately relies on the model’s ability to be able to correctly assess fairness and may also depend on the data used in pretraining and finetuning. While the experiments do suggest a positive impact of these techniques in mitigation, the authors can include a more diverse set of experiments to further boost confidence in their solution.

The authors call out that scrubbing of gender indicators and other protected information may not be enough to mitigate bias as such information may be still encoded in other ways such as gendercoded terms and may also result in loss of information. Having experiments that compare the effect of Scrubbing to FRL and TAICFRL could be useful in further cementing the efficacy of these  mitigation strategies.

Typo: page 22 “to perforn” -> “to perform”

---

### Meta-Review · Area_Chair_zj7L · 2026-02-26

**Recommendation:** Main Papers Track
**Confidence:** 3

**Metareview:**

The paper provides an interesting investigation into how memory modules in agentic systems can introduce and propagate biases in personalized hiring scenarios. The introduction of two novel strategies—Fairness Regulation Learning (FRL) and Task-Aware In-Context Fairness Regulation Learning (TA-ICFRL)—to nudge the model toward more equitable outcomes is well-motivated and demonstrates practical potential for AI safety and alignment.

**Key Feedback from Reviewers:**

* **Technical Quality:** Reviewers praised the structured approach to stage-wise bias decomposition and the novelty of addressing bias specifically within the memory-reading loop of agentic systems.
* **Limitations in Scope:** A primary limitation noted is the exclusive focus on **gender bias**. Since the evaluation relies on synthetic data, the framework's robustness would be significantly strengthened by including other protected attributes such as race, age, or country of origin.
* **Reliability:** Some concern was raised regarding the prompt-based nature of the mitigation strategies, as LLMs do not always guarantee strict compliance with such instructions.

**Conclusion:**
Overall, the results are interesting, and the methodology is solid. While the scope of bias types is narrow, the paper establishes a good step towards detecting "hidden" biases in personalized AI agents. The work is well-aligned with the workshop's goals and is recommended for acceptance.